# Novel Methodology for Localizing and Studying Insect Dorsal Rim Area Morphology in 2D and 3D

**DOI:** 10.3390/insects14080670

**Published:** 2023-07-28

**Authors:** Vun Wen Jie, Arttu Miettinen, Emily Baird

**Affiliations:** 1Department of Zoology, Stockholm University, 11418 Stockholm, Sweden; emily.baird@zoologi.su.se; 2Department of Physics, University of Jyvaskyla, 40014 Jyvaskyla, Finland; arttu.i.miettinen@jyu.fi; 3Swiss Light Source, Paul Scherrer Institute, 5232 Villigen, Switzerland

**Keywords:** allometry, bumblebee, dorsal rim area, micro-computed tomography, ommatidia, virtual histology

## Abstract

**Simple Summary:**

Insects can perceive and use polarized light in the sky when navigating by using a special region in their eyes called the dorsal rim area (DRA). The size and characteristics of the DRA can differ among insect species but, until now, there has been no effective method to investigate and compare these differences. We used a combination of 2D photography and X-ray imaging to determine the area of the DRA and demonstrated this method in the buff-tailed bumblebee. We showed that the DRA could be reliably obtained using 2D photographs and we found that the bumblebee body size affected the DRA. Using X-ray imaging, we also showed that the crystalline cones, which are light-focusing structures, were smaller in the DRA compared to the rest of the bumblebee eye. Our method was time-efficient, non-destructive, and could be used to determine photoreceptor characteristics at an individual level. This method will allow for more in-depth studies of the DRA in insects.

**Abstract:**

Polarized light-based navigation in insects is facilitated by a polarization-sensitive part of the eye, the dorsal rim area (DRA). Existing methods to study the anatomy of the DRA are destructive and time-consuming. We presented a novel method for DRA localization, dissection, and measurement using 3D volumetric images from X-ray micro-computed tomography in combination with 2D photographs. Applying the method on size-polymorphic buff-tailed bumblebees, *Bombus terrestris*, we found that the DRA was easily obtainable from photographs of the dorsal eye region. Allometric analysis of the DRA in relation to body size in *B. terrestris* showed that it increased with the body size but not at the same rate. By localizing the DRA of individual bumblebees, we could also perform individual-level descriptions and inter-individual comparisons between the ommatidial structures (lens, crystalline cones, rhabdoms) of three different eye regions (DRA, non-DRA, proximate to DRA). One feature distinct to the bumblebee DRA was the smaller dimension of the crystalline cones in comparison to other regions of the eye. Using our novel methodology, we provide the first individual-level description of DRA ommatidial features and a comparison of how the DRA varies with body size in bumblebees.

## 1. Introduction

Navigation is pivotal for many animals for their day-to-day tasks of foraging, finding mates, avoiding competition or predation, and returning home. While on such journeys, insects maintain their bearings using directional information acquired from the position of the sun [1,2,3], moon and stars [4,5,6], [7] (pp. 45–144), and [8], the spectral light distribution across the sky [9] (pp. 145–185) and landmarks [7] (pp. 45–144) and [10,11,12,13,14,15,16,17,18,19,20]. Insects can also obtain compass information using the pattern of polarization that is created in the sky [9] (pp. 145–185), [21] (pp. 291–348), and [22] by the scattering of sunlight in the Earth’s atmosphere [23] and [24] (pp. 23–56). While this pattern is invisible to us, it can be perceived by insects through a small dorsal subsection of their compound eyes called the dorsal rim area (DRA). The DRA ommatidia—the single visual units, or facets of a compound eye—are structurally different from those in the non-DRA regions and contain orthogonally arranged microvilli that enable the perception of polarized light [25,26,27].

The function of the DRA has been primarily studied by analyzing its structural morphology, which has now been detailed in multiple insect orders: Blattodea [28], Coleoptera [28,29,30,31,32,33,34], Diptera [28,35,36,37,38,39], Hymenoptera [40], [41] (pp. 41–60), and [42,43,44,45,46,47,48,49,50,51,52,53], Lepidoptera [54,55,56,57,58], Odonata [59], and Orthoptera [60,61,62]. Though these studies describe the DRA morphology of a particular species, they do not provide analyses of the variation between individuals (see [61] for an exception) or complete descriptions of each of the different ommatidial components within the DRA. This is due primarily to methodological limitations—the histological and electron microscopy methods typically used in these studies are destructive and are limited to non-isotropic analyses of the structures in either the longitudinal or the transverse plane in each specimen. To better understand how the DRA morphology varies both within and between species, and how this might ultimately relate to differences in functionality, methodologies that are capable of describing the ommatidial features within one specimen in both 2D and 3D are necessary.

To characterize the morphological features of the DRA, one must first localize and isolate it from the non-DRA regions of the compound eye. The localization of the DRA region can be conducted through the identification of orthogonally arranged microvilli [41] (pp. 41–60) and [47,52,57,59,60]. In species where it has been studied, the DRA is often visually distinct from other parts of the eye. For example, in locusts with light-coloured eyes, the DRA appears as a dark region, and in insects with black eyes, such as bumblebees, the DRA is often also indicated by a lighter region [34,40,46,47,61]. Here, we describe a new method that combines 2D photography with 3D data generated from micro-computed tomography (micro-CT) to localize and characterize the DRA in eyes where it is visually distinct. Our method began by superimposing a 2D image of the distinct DRA region of a compound eye onto the respective 3D reconstructed head model. This not only enabled the localization of the DRA region that is otherwise invisible in the 3D data but also facilitated detailed descriptions of the DRA morphology in individual specimens. In addition, our approach enabled an analysis of how the DRA ommatidia are distributed in 3D space, which can provide insights into how they sample the skylight polarization pattern, something that may have important consequences for their function within and between insect species. We applied our methodology to the bumblebee *Bombus terrestris*, which was chosen for this study because they have dark eyes with a grey dorsal region [41] (pp. 41–60) and they exhibit size-polymorphism [63] (pp. 21–43). We used our data to explore if and how the size of the DRA region in *B. terrestris* varied with both body and eye size. We demonstrated that we could accurately localize the DRA region by mapping the 2D images of individual eyes onto 3D volumetric scans of the same eyes and use this to guide virtual segmentation. Our approach was validated by using transmission electron microscopy (TEM) to compare the features of the virtually segmented DRA structures with features in other areas of the compound eye. Using our method, we provide the first complete description of the 3D morphological structures in DRA ommatidia in an individual insect and the first analysis of how the DRA surface area changes with eye size within a species.

## 2. Materials and Methods

### 2.1. Animal Handling

*Bombus terrestris* colonies were acquired from a commercial supplier (Koppert, Berkel, The Netherlands) and kept in incubators (Panasonic MIR Cooled Incubators, 123L, Tokyo, Japan) at 26 °C in complete darkness. They were provided with a 50% sugar water solution and fresh-frozen, organic pollen every two to three days (Naturprodukter, Raspowder Bipollen Ekologiskt bipollen). 

Twenty workers of different sizes were randomly selected and assigned individual ID numbers. The selected individuals were first anesthetized with CO_2_ and then sacrificed using ethyl acetate. The thorax was photographed (Canon IXUS 220 HS, 12.1) and the inter-tegular distance (ITD), the distance between the two insertion points of the wings that can be used as a proxy for body size [64], was measured in pixels and converted to mm using FIJI (Version 1.8.0_172) [65].

### 2.2. Acquisition of 2D DRA Images and Surface Area Measurements

A marked individual was placed in a 3D printed sample holder attached to the end of a micromanipulator (Figure 1A). The dorsal region of the left compound eye was viewed under a stereomicroscope with a camera attachment (Figure 1B). After removing the pile (soft hair) around the ocelli using micro-scissors, a small grid paper for size calibration was attached onto the midsection of the head. The grid was attached using super glue leaving at least one ocellus uncovered. The DRA was illuminated using a light source that was positioned at a 45° angle with respect to the DRA position (Figure 1C,D). The DRA and the grid paper were photographed (Canon EOS 70D) with a resolution of 1824 × 2432 pixels. The DRA surface area in 2D was calculated from the images using FIJI (Version 1.8.0_172) [65].

### 2.3. Staining and Embedding

The heads were dissected from the body and a microscalpel was used to remove the mouthparts. The heads were then immediately placed in 70% ethanol and 0.5% phosphotungstic acid (PTA) solution for 10 days [66]. The heads underwent dehydration using a graded ethanol series and were cleared in acetone prior to transferring them to epoxy resin (Agar 100). Samples in the wet resin were transferred onto a perspex block and cured in an oven at 60 °C for 48 h. After curing, excess external resin was removed to expose the eyes and surrounding cuticle [67]. 

For transmission electron microscopy (TEM), one *B. terrestris* was dissected and one of its compound eyes was fixed in paraformaldehyde (3%), glutaraldehyde, and glucose (2%), in a phosphate buffer for 1–3 h. The unstained sample underwent dehydration with a graded alcohol series and was embedded in epoxy resin (Agar100) which was cured at 60 °C for 2 days. After curing, the external resin was removed. A dorsal section containing the DRA and a ventral section of the compound eye that does not were cut. The two pieces were individually embedded and sliced to about 80 nm thick for TEM. The pieces were sliced transversally to the photoreceptors. The slices were first stained with Reynold’s lead citrate followed by uranyl acetate before imaging [68].

### 2.4. Micro-CT

Micro-CT scanning was performed at the TOMCAT beamline of the Swiss Light Source, Paul Scherrer Institute, Villigen (Switzerland) (beamtime number 20190641). Only 14 out of the selected 20 heads were scanned since the remaining 6 were damaged during staining and embedding. The intact heads were scanned using a monochromatic X-ray beam (20 keV); 2001 projection images of 2560 × 2160 pixels with a 60 ms exposure time were collected by a PCO.Edge 5.5 detector over 180° and a propagation distance of 100 mm. The scans were performed using a 4× objective. The projection images were processed with the Paganin phase-retrieval method [69] and reconstructed into 3D volumes using the gridrec algorithm [70] with Parzen filter, resulting in 16-bit volume images that had an isotropic voxel size of 1.625 µm. 

The regions of interest (ROI) from the reconstructions were cropped using the program Drishti Paint [71] and resaved as 16-bit TIFF files to reduce file size. Two ROI were obtained from the reconstructions of each sample, and they consisted of (i) the left compound eye (in the animal’s perspective), and (ii) the dorsal head (the dorsal part of the left compound eye and the ocelli, which were used as landmarks to facilitate mapping the 2D images in later analysis, Section 2.6).

### 2.5. The 3D Compound Eye and Surface Area Measuremets

The 3D models were generated from the cropped compound eyes in Amira (Version 2020.3.1, ThermoFisher Scientific, Waltham, MA, USA) using a combination of segmenting tools that included thresholding, brush, fill, shrink, and grow. The surface area of the compound eye was obtained by generating a surface from its segmented label [72].

### 2.6. Localizing the DRA on a 3D Head Model

From the 14 dorsal head volumes, six with the best-preserved eyes and high contrast micro-CT scans were chosen for the DRA localization and anatomical ommatidial analysis. We chose two samples from each of the three chosen body size ranges based on the ITD, small (2.965–3.902 mm), medium (3.902–4.839 mm), and large (4.839–5.776 mm). The 3D models of the eyes were generated in Amira using a combination of segmenting tools that included thresholding, brush, fill, shrink, and grow. Because the grey DRA region was not visible in the 3D volumes, the localization of the DRA of an individual bumblebee required both its 3D dorsal head and its 2D DRA image.

The DRA was localized by overlaying a 2D DRA image (Figure 1C) onto the 3D dorsal head volume taken from the same specimen (Figure 2A). The ocelli were used as landmarks for precise mapping (Figure 2B). Mapping was performed in the project view of Amira using modules that included the surface view and clipping plane. The DRA on the 3D model was segmented using the mapped image as a reference (Figure 2C). The surface area of the DRA was obtained by generating a surface using the segmented DRA labels.

To determine the accuracy of our DRA localization method, we took transverse sections of the rhabdoms in regions predicted to be DRA and non-DRA on a separate individual, and prepared and visualised them using transmission electron microscopy [68]. 

### 2.7. DRA Allometry and Overall Compound Eye Surface Area

Since the two methods used for determining the DRA surface area yielded very similar results (Figure 3), the surface area measurements of the DRA from the 2D photos were used for the allometric analysis of how DRA surface area varies with the compound eye surface area (*n* = 20). The surface area of the compound eyes was obtained from the 3D compound eye models (*n* = 14). All variables were converted to linear measurements before logarithmic transformation. The relationships between the surface area of the DRA and compound eyes were plotted against ITD onto a log–log plot.

### 2.8. DRA Ommatidial Structure Measurements

The compound eyes from the six individuals that had their DRA localized and segmented were also used for volumetric analysis. The three chosen regions for ommatidial structure comparisons were within the DRA, proximate to the DRA (five ommatidial rows surrounding the DRA), and non-DRA (>20th ommatidial row vertically from the DRA). Seven longitudinal virtual histological slices consisting of the three regions were obtained per eye from the 3D scans using the clipping plane module in Amira. Only virtual slices where the rhabdoms were continuous were used for further analysis (Figure 2C). Virtual histological slices were exported and analyzed in FIJI (Version 1.8.0_172) [65]. The thickness of the lens, crystalline cones, and rhabdoms, as well as the width of the crystalline cones were measured. The lens width was measured externally on the model using 3D line tools (Figure 2C). For each of the six eye samples, 20 ommatidial structure measurements were performed.

## 3. Results and Discussion

Our methodology allowed for the localization of the DRA on a homogeneous eye surface through the mapping of 2D images onto 3D volumetric data in individual compound eye samples and a comprehensive study of the gross internal and external morphology of the DRA structures. We discovered that the external 2D images were sufficient for calculating the DRA surface area, confirming that this is a valid method for more accessible high-throughput analyses of DRAs within and between species.

### 3.1. DRA Localization on a Homogeneous Eye Surface

This is the first localization of the DRA through the mapping of 2D images onto 3D volumetric data generated from micro-CT scans. We used TEM to analyze the microvilli of transverse sections of the DRA and non-DRA regions of the compound eye. The presence of orthogonally arranged microvilli in the region identified as the DRA suggests that our localization was accurate (Figure 4A,B). In addition, since the structures were relatively flat, the comparative analysis between these and the 2D images showed that there were no significant differences between the methods (t = −1.0421, *p* = 0.3451, *n* = 6, Wilcoxon signed-rank test) (Figure 3). This means that the 2D external images alone can provide an easy and high-throughput estimation of the DRA surface area. Although the combination of photos and micro-CT data is not new [73], we demonstrated here that it can also be used to make quantitative measurements. This methodology opens up new ways to study the DRA and other external features since it allows for individual-level gross morphological descriptions of these structures.

### 3.2. A Comprehensive Study of the Internal and External Morphology of DRA Structures

In contrast to previous methodology for characterising ommatidial structures in the DRA using TEM, our method allowed us to achieve gross morphological descriptions of most components of the DRA ommatidia within an individual insect due to its non-destructive nature. We found that the ommatidial structures were similar between the areas within and proximate to the DRA regions (Figure 4). The most distinct difference that we observed between the DRA and the other eye regions was the thickness of the crystalline cones; those in the DRA were both thinner (DRA vs. proximate to DRA: H_2_ = 10.211, *p* = 0.0223; DRA vs. non-DRA: H_2_ = 10.211, *p* = 0.0065, *n* = 6, pairwise Wilcoxon rank sum) and smaller in diameter (DRA vs. proximate to DRA, H_2_ = 13.93, *p* = 0.0087; DRA vs. non-DRA: H_2_ = 13.93, *p* = 0.0065, *n* = 6, pairwise Wilcoxon rank sum) than those in other regions. This is unlike what has been observed in other insects, such as the Canarian tiny cricket [60] and the desert locust [62]. The question arises to as why some insects have distinct differences between the DRA and the regions proximate to it while others do not. The comprehensiveness of this methodology allowed us to pick up minute differences between the regions of interest and showed that structures within the DRA and proximate to the DRA can be very similar, at least in *Bombus terrestris*.

### 3.3. Method Implementation on Bombus terrestris to Study How the DRA Varies with Body Size

This study provided an easy and high-throughput methodology for studying the DRA surface area in insects that allowed us to perform the first allometric study of how the surface area of the DRA scales with body size in bumblebees. Our results revealed that the DRA, as a small component of the compound eye, correlated positively with body size (t_12_ = 2.682, *p* = 0.0152, *n* = 20, regression analysis) (Figure 5). The compound eye also correlated positively with body size (t_12_ = 7.137, *p* < 0.001, *n* = 20, regression analysis). This result is consistent with the findings of Taylor et al. [72] and suggests that the variation of eye size with body size likely affects visual capabilities such as resolution and sensitivity in *B. terrestris*. This, in turn, is likely to affect visually guided behavior, such as the timing of activity [74,75]. Do the differences in the DRA observed here relate to functional differences, for example, in navigational capabilities (by affecting the sensitivity to polarised light in the DRA)? Our method allows questions like this to be answered by facilitating the quantification of the effect of body size on the size of the DRA and could ultimately inspire more questions about the DRA functionality that can be tested behaviourally.

### 3.4. Limitations of the Method

While our method provided new insights into the morphological structures of the DRA, both externally and internally within one individual, it had some limitations that are important to highlight. Although micro-CT based 3D volume renderings of compound eyes allow for a more comprehensive gross morphological description of the DRA structure, they do not have the resolution necessary for resolving the orientation of individual microvilli. This limitation should be addressed in the future as microvilli orientations within the DRA determine how the polarisation pattern of the sky is sampled. However, as the samples prepared for micro-CT are embedded in resin and stained with heavy metal, they are also appropriate for performing TEM [76]. This correlative approach would allow for the mapping of the microvilli arrangement at the individual level that, to our knowledge, has never been achieved in 3D. A further limitation of our method was that the 3D methodology took a long time to learn, required advanced amounts of computing power (gigabytes of RAM), storage space (data range in gigabytes per dataset), and could be expensive. The 2D methodology, though it is considered high throughput, only works if the DRA structures are relatively flat, which is the case in *B*. *terrestris*. This is not the case in butterflies as their DRAs are curved (ranging from the edge of the eye next to the antennae to the dorsal edge of the eye) [56]. To acquire the surface measurements of these, the 3D localization method is necessary. Finally, this method is limited to analyzing the DRAs of insects with a pronounced DRA that is visible in 2D photographs.

## 4. Conclusions

To understand insect navigational behaviour, it is necessary to study all sensory structures involved, including the DRA that is linked to polarization vision-based navigation. Though this structure is widespread among insects, it has proven to be problematic to study, due to the lack of complete and non-destructive individual-level methods. Our novel methodology presented in this study allowed us to produce more insightful observations of the DRA morphology and, in turn, raise questions about polarization vision-based navigation in insects.

## Figures and Tables

**Figure 1 insects-14-00670-f001:**
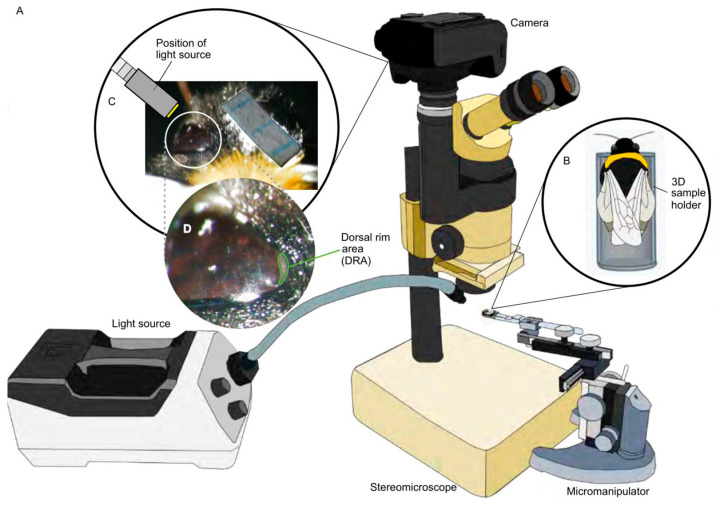
Set-up and equipment for the dorsal rim area (DRA) image acquisition in 2D. (**A**) microscope set-up to acquire DRA photographs, (**B**) bumblebee in sample holder, (**C**) light set-up for DRA visualization, (**D**) enlarged view of the DRA.

**Figure 2 insects-14-00670-f002:**
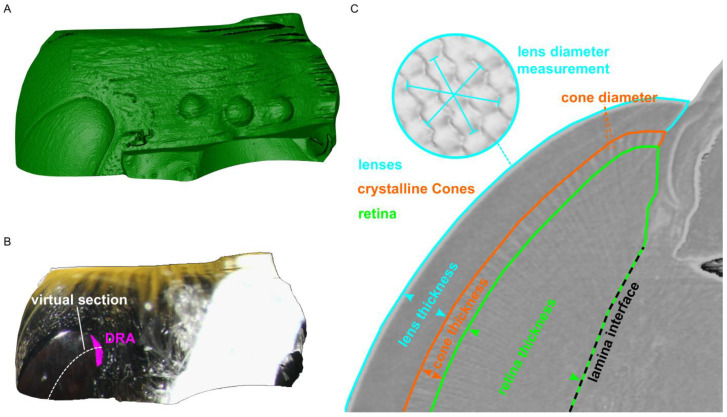
The process for localizing the DRA on a 3D head model of a bumblebee *Bombus terrestris*. (**A**) A 3D volume rendering of a bumblebee head. (**B**) A 2D photo is overlayed onto the 3D volume and is used to identify and segment the DRA (purple outline). The white dashed line indicates the region of the virtual slice (**C**) that shows the ommatidial structure and how the different ommatidial measurements were taken.

**Figure 3 insects-14-00670-f003:**
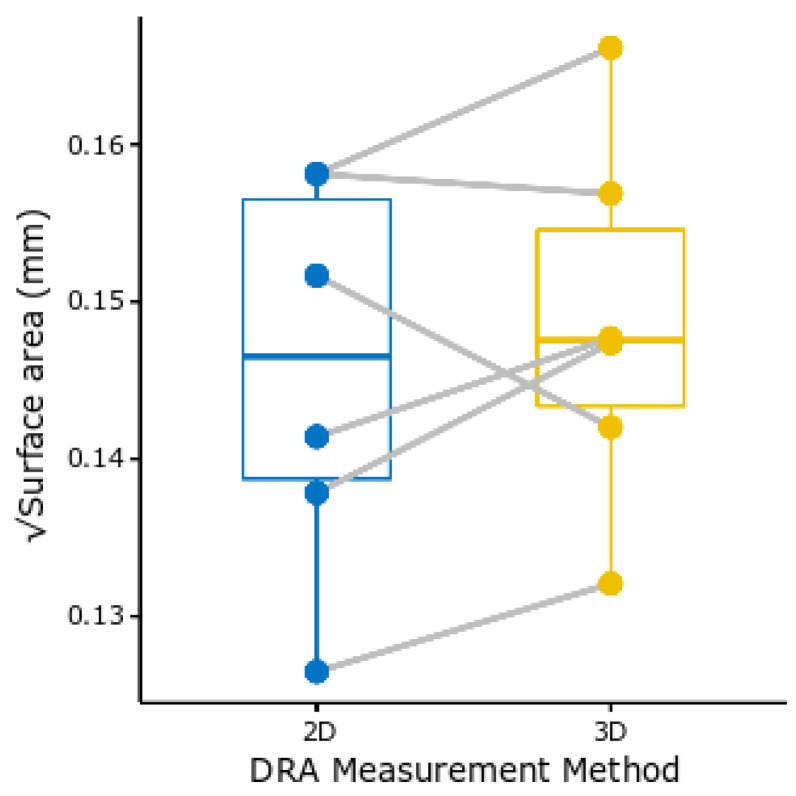
Boxplot comparing 2D and 3D DRA surface area measurements taken from the same individuals (grey lines) of *Bombus terrestris*. There was no significant difference between the values acquired using the 2D images or 3D volumes (t = −1.0421, *p* = 0.3451, *n* = 6, Wilcoxon signed-rank test).

**Figure 4 insects-14-00670-f004:**
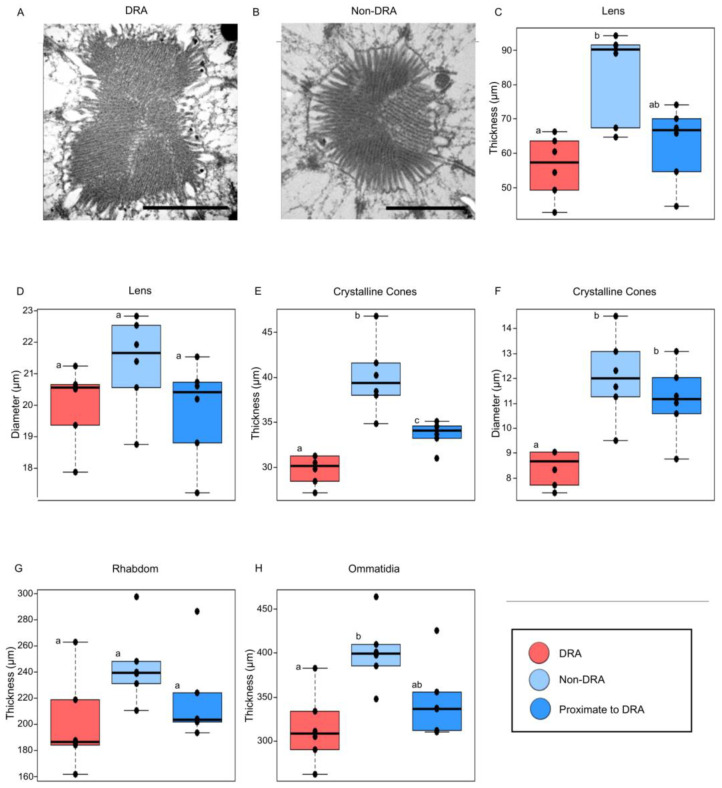
Transverse sections of (**A**) DRA, and (**B**) non-DRA microvilli obtained from transmission electron microscopy (TEM) (*n* = 1). Ommatidial structure comparison between DRA, proximate to the DRA and non-DRA regions. Thickness of lens (**C**), crystalline cones (**E**), rhabdom (**G**), and ommatidia (**H**). Diameter of lens (**D**), and crystalline cones (**F**) (*n* = 6). Letters represent the outcome of pairwise Wilcoxon rank sum tests, matching letters indicate no significant difference at *p* ≥ 0.05 and non-matching letters indicate significance below this value. Scalebar: 1 μm.

**Figure 5 insects-14-00670-f005:**
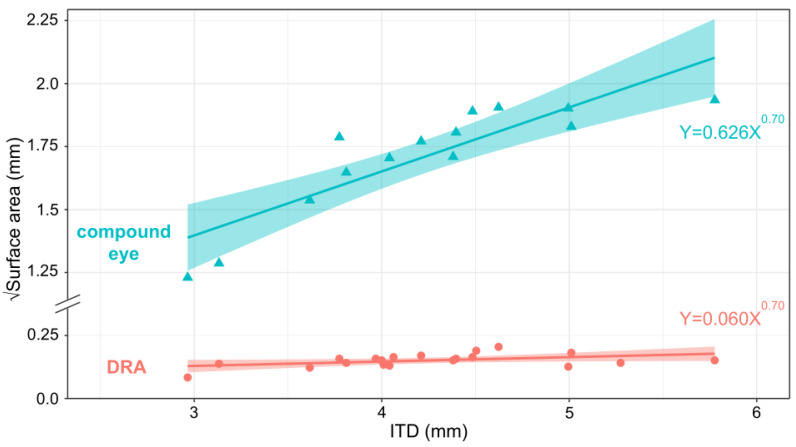
Allometric relationship between the square root of eye surface area (mm) and ITD (mm). Triangles = 3D compound eye measurements (t_12_ = 2.682, *p* = 0.0152, *n* = 14, regression analysis), circles = 2D DRA measurements (t_12_ = 7.137, *p* < 0.001, *n* = 20, regression analysis).

## Data Availability

Upon publication, raw data supporting the findings in this paper will be made available on Figshare (https://doi.org/10.6084/m9.figshare.23628657) accessed on 5 July 2023. For review purposes, the data can be accessed through a private link (https://figshare.com/s/6f940547107ae3931d40 accessible till 1 December 2023). The samples used in this study are stored at the Department of Zoology, Stockholm University, Sweden.

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
