# Peer review of "Novel Methodology for Localizing and Studying Insect Dorsal Rim Area Morphology in 2D and 3D"

_insects, 2023, doi:10.3390/insects14080670_

Round 1
Reviewer 1 Report
I really like this manuscript! Tangible scientific progress regarding both, methodology (mapping of DRA region on an individual level in bumblebee) and actual description of DRA allometry in the bumblebee. This method and its initial application may well open up a plethora of studies on insect navigation. The text is concise, clear, complete, no deficits I could pinpoint (except really minor notes below). And it is fair in considering potential limitations, very nice (and nowadays not all that common).
Minor comments.
1. The relevant parts - DRA & head capsule - in Fig. 1C, D are too dark to be usefully discernible. Adjust brightness / contrast?
2. Labelling text in Fig. 2D, green and orange, is not readable. Larger font or black lettering with green and orange underlining?
3. You provide rather more than fewer references, which is alright (not least regarding potential reviewers (;-)). Reference 11 is not on landmark use , however. Maybe you mixed it up with Wolf & Wehner (2000) Pinpointing food sources... J Exp Biol 203: 857-868?
The English text is perfect throughout, could not pinpoint any errors, tyopos to grammar. The only exceptions are Simple Summary and Abstract (perhaps added in a hurry in the very end)?
line 11, ... rim area (DRA).
14, buff-tailed
18-19,time-efficient
20, a suggestion: This... instead of An ...
24, it is not just localization, is it?
26, size-polymorphic
30, delete comma
Reviewer 2 Report
The paper is scientifically sound and interesting, there are perhaps some minor problems I evidenced on the PDF text with comments. The reader could be intrigued by the use of TEM in the two photographs of figure 4 (A, B), which lack scale and where the method is not indicated in the caption. The authors should add method details here and the scale, moreover the resolution of the two figures is low, I argue it is a problem of the draft (or not), but in my opinion the quality of A & B should improved. In the methods it should be indicated that the staining/embedding procedure of microtomography and TEM was the same, not only in the discussion. In fact, there are other x-ray scanning mehods in which such preparation is not necessary (eg in synchrotron phase contrast microtomography), not all readers are so well informed in such methods.
With such small additions the paper is in my opinion acceptable.
